# Role of Neuroinflammation and Blood-Brain Barrier Permutability on Migraine

**DOI:** 10.3390/ijms22168929

**Published:** 2021-08-19

**Authors:** Gaku Yamanaka, Shinji Suzuki, Natsumi Morishita, Mika Takeshita, Kanako Kanou, Tomoko Takamatsu, Shunsuke Suzuki, Shinichiro Morichi, Yusuke Watanabe, Yu Ishida, Soken Go, Shingo Oana, Yasuyo Kashiwagi, Hisashi Kawashima

**Affiliations:** Department of Pediatrics and Adolescent Medicine, Tokyo Medical University, Tokyo 160-0023, Japan; shin.szk@gmail.com (S.S.); sunflowernk69@gmail.com (N.M.); jerryfish_mika@yahoo.co.jp (M.T.); kanako.hayashi.0110@gmail.com (K.K.); t-mori@tokyo-med.ac.jp (T.T.); suzu_shun_0705@yahoo.co.jp (S.S.); s.morichi@gmail.com (S.M.); vandersar_0301@yahoo.co.jp (Y.W.); ishiyu@tokyo-med.ac.jp (Y.I.); soupei59@gmail.com (S.G.); oanas@tokyo-med.ac.jp (S.O.); hoyohoyo@tokyo-med.ac.jp (Y.K.); hisashi@tokyo-med.ac.jp (H.K.)

**Keywords:** blood-brain barrier, migraine, neuroinflammation, IL-1β, chemokine, anakinra

## Abstract

Currently, migraine is treated mainly by targeting calcitonin gene-related peptides, although the efficacy of this method is limited and new treatment strategies are desired. Neuroinflammation has been implicated in the pathogenesis of migraine. In patients with migraine, peripheral levels of pro-inflammatory cytokines, such as interleukin-1β (IL-1β) and tumor necrosis factor-α, are known to be increased. Additionally, animal models of headache have demonstrated that immunological responses associated with cytokines are involved in the pathogenesis of migraine. Furthermore, these inflammatory mediators might alter the function of tight junctions in brain vascular endothelial cells in animal models, but not in human patients. Based on clinical findings showing elevated IL-1β, and experimental findings involving IL-1β and both the peripheral trigeminal ganglion and central trigeminal vascular pathways, regulation of the Il-1β/IL-1 receptor type 1 axis might lead to new treatments for migraine. However, the integrity of the blood-brain barrier is not expected to be affected during attacks in patients with migraine.

## 1. Introduction

Migraine is now the sixth most common disease globally [1] and one of the three neurological disorders with the highest absolute burden of disability-adjusted life-years in the United States [2]. With a better understanding of the neurobiology of migraine and the 5-hydroxytryptamine (5-HT) receptor, or serotonin receptor, the burden on patients with migraine has been lightened with the creation of selective 5-HT_1B/1D_ agonists called triptans [3]. However, migraine still imposes a significant burden on individuals and society from childhood to adulthood. An important issue in the study of migraine is the identification of neurotransmitters and neuromodulators that may be involved in the pathophysiology of migraine [4].

Currently, the dominant role of the calcitonin gene-related peptide (CGRP) pathway is the main focus of migraine research and has been validated and clinically applied by a series of translational studies [5]. An overview of the pathophysiology of migraine is shown in Figure 1, with an emphasis on the role of CGRP in the trigemino-vascular system. However, although CGRP has also been implicated in the activation of inflammatory cells, there is a limit to the effectiveness of CGRP-targeted therapy [6]. As novel treatment strategies are being explored, the concept of neuroinflammation could be a productive option to investigate.

Cortical spreading depression (CSD), relevant to the development of migraine aura, has been shown to activate the trigemino–vascular system (TVS). Anti-temporal conduction of trigeminal ganglion neurons leads to the release of neuropeptides, including calcitonin gene-related peptide (CGRP), from their nerve terminals, leading to vasodilation and plasma extravasation. The dura is densely innervated by sensory nerve fibers that contain these neuropeptides originating from the trigeminal ganglion (TG), causing neurogenic inflammation in the dura (A). In contrast, direct conduction of TG neurons generates a pain sensation via the activation of c-fos in trigeminal nucleus caudalis (TNC) (B), which is eventually perceived as a headache. CGRP receptors are observed in TGs and TNCs but not in peripheral trigeminal nerve terminals, suggesting that CGRP receptor antagonists may suppress neurons in TGs and TNCs. An explanation of how CSD stimulates TVS is given in Figure 1.

Neurogenic inflammation in migraine is mainly characterized by the release of neuropeptides such as CGRP and substance P from the trigeminal nerve, leading to arterial vasodilation, plasma protein extravasation, and mast cell degranulation. The involvement of these neuropeptides in migraine is evident [7,8], and pro-inflammatory cytokines or chemokines may be involved in this series of reactions [9,10].

Neuroinflammation is essentially defined as an inflammatory response in the brain and spinal cord, which is modulated by the production of cytokines, chemokines, reactive oxygen species (ROS), and secondary messengers such as nitric oxide (NO) and prostaglandins [11,12]. They are mainly produced by activated microglia and astrocytes [11,12], and neurovascular units comprising neurons, pericytes, and endothelial cells [13,14]. Neuroinflammation is closely related to central nervous system (CNS) diseases, such as multiple sclerosis and epilepsy, and the regulation of cytokines has revealed new therapeutic aspects.

This review explores the current literature on neuroinflammation. It targets neuroinflammation from the perspective of cytokines in the pathogenesis of migraine and aims to identify new directions for research on therapeutic interventions for migraine targeting neuroinflammation.

## 2. Summary of Search

This narrative literature review is based on studies on the involvement of oxidative stress, neuroinflammation, and mitochondrial dysfunction in migraine, and the effectiveness of riboflavin as a prophylactic treatment for migraine. A literature search was conducted in the PubMed database and included articles published up to June 2021. The keywords used in the search were “migraine” and “inflammation” or “neuroinflammation” or “cytokine” or “chemokine” or “blood brain barrier.”

## 3. Clinical Evidence of Neuroinflammation

There have been a number of studies on cytokines and chemokines in patients with migraine, including studies performed during the attack, in the interictal period, and even during the post-attack period.

During migraine attacks (ictal period), interleukin-1β (IL-1β), IL-6, IL-8, and tumor necrosis factor-α (TNF-α) are increased [15,16,17,18,19,20]. Pediatric patients without aura showed higher levels of IL-1β than those with aura [17]. However, some reports have shown no significant difference in TNF-α and IL-1β levels between attacks and attack-free periods [21,22]. Decreased levels of the anti-inflammatory cytokines IL-4 and IL-5 were reported during the attacks [18,23]. Increased levels of IL-10 (an anti-inflammatory cytokine) during attacks, compared with the interictal period [16,24,25], and a decrease in IL-10 by sumatriptan (one of the triptans) were found [25].

During headache-free periods (interictal period), increased pro-inflammatory levels of IL-1β, IL-6, TNF-α, and chemokines such as IL-8, macrophage inflammatory protein-1α (MIP-1α), and C–C chemokine ligand 5 (CCL5) were observed in patients [24,26,27,28,29]. Conversely, IL-10 levels were found to be similar [15] or decreased [25,26,27] when compared with healthy individuals.

Studies measuring interictal cytokine concentrations in patients with migraine have reported conflicting results, with some showing no differences between migraine and control groups for TNF-α and soluble TNF receptors TNF-R1 and -R2 [22,30], while others show increased levels in patients with migraine [26]. IL-6 and IL-8 levels were lower than those in the control group [28].

Some reports have examined the association between these markers, headache severity, and comorbidity. Plasma TNF-α is negatively correlated with anxiety scores [28], and the serum MIP-1 or MIP-1:CD14 ratio was reported to be associated with the pain index and the severity of headache [31]. However, elevated serum IL-8 and MIP-1α do not appear to be associated with comorbid psychiatric disorders or allodynia [29].

### 3.1. Serial Analysis of Cytokines

While there is some directionality, such as an increase in pro-inflammatory cytokines during both migraine attacks and interictal periods, no definite consensus has been reached. The duration of headache attacks varies widely, which may explain the inconsistent results in terms of sampling timing. Serial analysis of cytokines from jugular venous blood by Sarchielli et al. yielded consistent and robust data which, quite intriguingly, were completely different from data taken from the periphery [32,33]. Significant increases in TNF-α, IL-6, nuclear factor (NF)-κB, and soluble intercellular adhesion molecule-1 (sICAM-1) were observed in parallel within 2 h of attack onset compared with the time of catheter insertion [33]. CGRP increased significantly after 1 h and IL-8 reached its highest level at 4 h, while the other two chemokines, RANTES (CCL5) and monocyte chemoattractant protein-1 (MCP-1), did not significantly change at any time point [32]. While a slight increase in IL-1β was observed from 1 to 4 h, the levels of IL-1β decreased, reaching values at the end of the attack [18]. The levels of cytokines, selectins, vascular adhesion molecules, and adhesion molecules in the peripheral blood of patients with migraine were found to be unaltered at each time point of the study [32,33]. The results of these investigations are consistent and significant over time, illustrating the difficulty of examining cytokines in patients with migraine. Carotid artery blood may provide clearer information than peripheral blood regarding biochemical and neurotransmitter changes in the cerebral circulation, and may be more characteristic of migraine attacks.

### 3.2. Cerebral Spinal Fluid (CSF)

Despite the normal serum TNF-α levels in patients with chronic migraine (CM), elevated TNF-α levels in the CSF have been identified [34]. Although IL-1 receptor antagonist (IL-1RA), MCP-1, and transforming growth factor-β1 were significantly increased in the CSF of patients with episodic tension-type headache and in migraine patients with and without aura compared to those without pain, the increases were not sufficiently different between these headache types [35]. A comparison of CSF and plasma or serum cytokine levels would provide some insight into the origin of cytokines, although the assessment of CSF cytokines in patients with migraine is limited. Due to ethical issues, it may be difficult to examine CSF in patients with migraine for the foreseeable future, hence the value of studying available reports.

### 3.3. Genetic Analysis

Consistent with the predominant theory of migraine that emphasizes vascular etiology, a genome-wide association showed that loci associated with migraine are rich in single nucleotide polymorphisms in vascular and smooth muscle tissues [36]. However, the frequency of the TNF-α-308 GG genotype was lower in the patient group than in the control group [37], suggesting a link between migraine and inflammation. In addition, current genetic analysis of muscle biopsies in the calvarial periosteum from patients with CM (where the pain was situated) found the expression of pro-inflammatory genes (e.g., CCL8 and toll-like receptor 2) to be significantly increased in patients with CM, attesting to muscle tenderness, whereas the expression of genes that suppress inflammation and immune cell differentiation (e.g., IL-10 receptor subunit alpha and colony-stimulating factor 1 receptor) was decreased [38]. These results might reflect the activation of the neuroinflammatory cascade in migraine pathogenesis.

## 4. Experimental Evidence of Neuroinflammation

To investigate the relationship between migraine and neuroinflammation, two main models, the cortical spreading depolarization (CSD) mouse model and the nitroglycerin (NTG)-induced rat model, were used, and the findings from each model are as follows.

### 4.1. Inflammation Response Provoked by CSD

CSD is recognized as a characteristic feature of migraine with aura, representing a strong wave of neuronal depolarization with glial and vascular activation [39,40], which was originally published by Leao [41]. The clinical significance of CSD, which can occur after ischemic, hemorrhagic, or traumatic brain injury, has been emphasized and recognized [42,43]. In experimental animals, CSD can activate trigeminal nociceptors [44] and is inhibited by drugs used for migraine prophylaxis [45]. The mouse model of CSD was used as an experimental model relevant to the pathophysiology of migraine.

CSD can trigger a substantial inflammatory response via the opening of neuronal Pannexin 1 megachannels and activation of caspase-1, followed by the release of high-mobility group box 1 from neurons and the activation of astrocytes [46]. Furthermore, the development of CSD has been shown to cause inflammation of the meninges by activating macrophages and mast cells and enhancing the production of several inflammatory cytokines, such as IL-1β, IL-6, and TNF-α [47,48,49,50,51].

In vitro analysis of spreading depression in hippocampal organotypic cultures [52] and astrocytes [51] reported upregulation of pro-inflammatory cytokines, such as IL-6, IL-1β, and TNF-α. Additionally, increased levels of IL-1β, IL-6, and TNF-α were detected in the representative studies of in vivo rodent models [49,53,54]. However, some studies could not detect similar findings [55,56].

These conflicting and inconsistent results might be due to the invasive method of CSD induction, which might induce an inflammatory response [55]. In transgenic mice expressing channelrhodopsin-2 in neurons (Thy1-ChR2 YFP transgenic mice) [57,58], optogenetics enabled an almost non-invasive induction of CSD [59,60]. Takizawa et al. demonstrated that non-invasive induction of CSD also evoked a potent pro-inflammatory response in mice, such as cytokines IL-1β, TNF-α, IL-6, chemokine CCL2, and cell adhesion molecules (ICAM-1 and vascular cell adhesion molecule-1) after a single cell or multiple CSDs [60]. Interestingly, these responses were attenuated in IL-1 receptor type 1 (IL-1R1) knockout mice, suggesting the involvement of IL-1β as an upstream mediator [60]. However, the inflammatory response induced by CSD was suppressed by dexamethasone but not by ibuprofen, the most commonly used acute migraine medication [60]. Ibuprofen, an anti-inflammatory agent, might act downstream after an inflammatory response, although a discrepancy between clinical practice and basic research appears to exist. The efficacy of dexamethasone in acute migraine is reported to have a negative effect on improvement [61] and short-term prevention [62,63] of migraine headaches. The suppression of the inflammatory response by dexamethasone is a recognized fact, and there is limited evidence of clinical findings with dexamethasone. There is no doubt that there is concrete evidence of neuroinflammation caused by CSD, and the suppression of inflammation by CSD might not necessarily translate into migraine treatment.

### 4.2. Spontaneous Migraine-like Mouse Model Using NTG

NTG administration can induce spontaneous migraine-like headaches in rodents. NTG is a vasodilator that acts as a nitric oxide donor and reproduces multiple symptoms of migraine, inducing hyperalgesia at the trigeminal and spinal levels [64].

In a rat model of meningeal inflammation following NTG, pro-inflammatory cytokine IL-1β increased in the dura mater and IL-6 increased in dural macrophages and the CSF, while NOS mRNA up-regulation was observed within the first hours after the infusion [65,66]. The increased transcriptional activity of inflammatory cytokines (IL-6 and TNF-α), CGRP, and nNOS with NTG-induced hyperalgesia was also detected in the trigeminal ganglia, cervical spinal cord, and medulla-pons, and blocked by the injection of kynurenic acid analog 1, an endogenous regulator of glutamate activity [67].

In addition, the peripheral inhibition of fatty acid amide hydrolase, an enzyme that deactivates the endocannabinoid anandamide, can lead to an antihyperalgesic effect with a decrease in CGRP, substance P, and TNF-α and IL-6 mRNA in two complementary rat models of episodic CM [68].

In the central area of the trigeminovascular pathway, activation of the microglial leucine-rich repeat pyrin containing protein-3 (NLRP3) inflammasome mediates the release of IL-1β and contributes to central sensitization in the NTG-induced migraine model. Additionally, NTG-induced CM-like pain was paralleled by activation of the NLRP3 inflammasome [69]. High levels of extracellular K^+^ ions, mitochondrial dysfunction, and the production of ROS are also known to activate NLRP3, which has been shown to underlie the development of migraine [70,71,72,73]. MCC950, an inhibitor of NLRP3 inflammasome, might also have a preventive effect on migraine occurrence.

## 5. Presence of Blood-Brain Barrier (BBB) Permutability in Migraine

The presence or absence of BBB impairment associated with neuroinflammation is important when considering therapeutic strategies. Studying a drug’s permeability from the periphery to the brain, and its potential to infiltrate cells such as monocytes that introduce cytokines into the brain, will lead to new therapeutic strategies in the future.

### 5.1. Experimental Research of BBB Permutability

Experimental studies of rat models resembling primary headache have shown that CSD modulates the permeability of the BBB by activating matrix metalloproteinase-9 (MMP-9) from 3 h after CSD induction, peaking at 24 h [74]. A recent study in awake rats also found that cortical BBB leakage began 0.5 h after CSD induction and resolved within 6 h, without altering the tight junction proteins occludin or claudin-5 [75]. CSD-induced BBB opening to water and large molecules is mediated by increased endothelial transcytosis. These phenomena are dependent on caveolin-1 and rho-kinase 2; in contrast, endothelial tight junctions, pericytes, and basement membranes are maintained after CSD [76]. This change in BBB permeability is not destructive and may be a transient alteration associated with an inflammatory response. It may also directly affect the access of agents to centrally located targets during migraine attacks [77]. Indeed, increased uptake of sumatriptan into the brain was detected, in association with transient BBB permeability, in a KCl-induced model of episodic headache [75]. Previous reports suggest that dural inflammation induced by trigeminal ganglion (TG) stimulation does not affect the integrity of the BBB [78].

### 5.2. Clinical Studies of BBB Permutability

Previous clinical studies have found no association between primary headache and BBB opening [79]. Elevations of MMP-9 [80] and ICAM-1 [19], which are considered typical markers suggestive of BBB disorders, are observed in patients with migraine. An MMP-9 haplotype was reported to affect circulating MMP-9 levels in women with migraine [81].

One study reported a significant elevation of endothelial cell-specific molecule-1 (ESM-1) and claudin-5 in the migraine attack group [20]. ESM-1 is expressed in the vascular endothelium and is regulated by a number of cytokines, including IL-1β and TNF-α, and growth factors. Claudin-5 regulates BBB permeability and is important in sustaining the integrity of cerebrovascular endothelial cells [82,83]. A positive correlation was also found between Visual Analog Scale scores and the levels of ESM-1 and claudin-5 [20]. In addition to the association of BBB disorders with the pathogenesis of migraine, the results suggest a link between the clinical severity of migraine and BBB disorders.

However, several recent magnetic resonance imaging (MRI) studies also demonstrated no increased BBB permeability during spontaneous migraine attacks with [84] and without aura [85]. These studies were performed relatively early, between 6.5 and 7.6 h from attack onset to MRI scanning [84,85], with reference to rodent studies [74]. In addition, glyceryl trinitrate-induced dural inflammation could not affect BBB function during ictal and interictal periods, and the passage of dihydroergotamine through the BBB was also negative [86]. These clinical results indicate that the integrity of the BBB is unlikely to be affected during a migraine attack, contrary to the results from animal models.

## 6. Participation of Pericytes in Migraine

Pericytes are a neurovascular unit component of the BBB and play an important role in the integrity of the BBB. Pericyte degeneration and/or dysfunction contribute to the loss of BBB integrity, which is an early hallmark of several neurodegenerative and inflammatory conditions [13,87,88].

Capillary pericytes are also shown to play an active role in the regulation of cortical vasculature during and after CSD [89]. In fact, prolonged vasoconstriction caused by CSD was revealed to be strongest in primary capillaries, where pericytes have a sustained increase in calcium levels. Somatosensory stimulation after CSD causes no further changes in the diameter of capillaries or calcium in pericytes, suggesting that pericytes play a critical role in long-term oligemia after CSD [89]. Current research has demonstrated that brain pericytes respond to inflammatory signals, including IL-1β and TNF-α [87,90,91,92,93]. Pericytes may act as sensors for the inflammatory response in the CNS [92,94]. Based on BBB integrity and systemic peripheral inflammation, pericytes are speculated to play a pivotal role in the pathogenesis of epilepsy during neuroinflammation [14]. Structural changes in pericytes associated with epileptic seizures are exacerbated by IL-1β rather than TNF-α [95]. It is uncertain whether migraine, like epilepsy, is associated with BBB disorder and neuroinflammation, and experimental results targeting pericytes have not been obtained. However, structural changes in pericytes and the endothelial cells of microvessels were also observed in patients with familial hemiplegic migraine (FHM) [96]. It is known that both migraine and epilepsy share pathologies and clinical features to a certain extent, through similar underlying pathophysiological mechanisms, and antiepileptics such as valproic acid or topiramate are effective in treating patients with both disorders [97,98]. Therefore, the prevention or treatment of pericyte constriction may become a therapeutic target in migraine [89,99].

## 7. Prospect of Therapeutic Benefit via the IL-1β/IL-1R1 Axis

IL-1β is a master cytokine that drives both brain and systemic inflammation, which activates a ubiquitous cell surface receptor, IL-1R1. The IL-1β/IL-1R1 axis activates a cascade of inflammatory molecules, including other cytokines and chemokines. In vivo analysis of meningeal nociceptors in the TG demonstrated that IL-1β activates and sensitizes meningeal nociceptors [100] and has been proposed as a key mediator in trigeminal activation after CSD [46]. In addition to the experimental models described above, increased gene expressions of IL-1β and IL-1RA were also detected in a mouse model of FHM type 1 (FHM-1) [101,102]. IL-1β mediates the pro-inflammatory process, promotes the activation of trigeminal satellite cells, and supports the cross-excitation of satellite glial cells and neurons in the TG [11,103].

IL-1 also plays an important role in the pathogenesis of epilepsy, and its antagonist, IL-1RA, exhibits brain-protective effects as well as potent anticonvulsant effects [104,105]. Anakinra, a human recombinant version of IL-1RA, has emerged as a new treatment option for intractable epilepsy and exerts its neuroprotective effects by suppressing epileptic seizures in fairly intractable epilepsy [106,107,108].

The usefulness of anakinra was also suggested for CNS diseases such as stroke [109] and subarachnoid hemorrhage [110]. Interestingly, anakinra can improve headache as well with the resolution of CNS inflammation in patients with cryopyrin-associated periodic fever syndromes—a spectrum of rare inherited autoinflammatory syndromes [111,112]. Based on these facts, we should also expect anakinra to be effective in treating migraine. However, in conditions such as migraine where the BBB is not damaged or is only compromised to a limited extent, unlike epilepsy, there is some concern that anakinra may have difficulty reaching the brain [113,114]. Anakinra was found to cross the BBB in a dose-dependent manner in nonhuman primates [115]; in fact, a dose-dependent effect was observed in patients with epilepsy [108].

The inhibitory effect of anakinra on neuroinflammation has been reported not only in the CNS but also in the peripheral nervous system. In mouse models of familial amyloid polyneuropathy, anakinra inhibits transthyretin expression in Schwann cells [116] and extracellular deposition, and protects against unmyelinated fibrillary degeneration [117].

In experimental models, along the course of the IL-1β/IL-1R1 axis, inhibition of the NLRP3 inflammasome by MCC950, but not by anakinra, led to suppressed NTG-induced hyperalgesia and decreased protein expression levels of c-fos and CGRP in the trigeminal nucleus caudalis. MCC950 is the most advanced NLRP3 small molecule inhibitor [118], and there might be an option other than anakinra for IL-1β suppression through NLRP3 inflammasome inhibition by MCC950 [69] (Figure 2).

Recently, curcumin, a dietary polyphenol, has been demonstrated to significantly inhibit IL-1β and TNF-α expression, oxidative stress, and protein accumulation, and to eventually protect neurons from neurodegeneration in several different disease models [119,120,121,122]. Additionally, curcumin has been shown to have a preemptive analgesic and antioxidant effect on experimental migraine [123].

CSD can trigger Pannexin-1 megachannel opening and mediate the formation of the inflammasome, comprising apoptosis-associated speck-like protein (ASC), NLRP3, and pro-caspase-1, in neurons of the cortex area. Pro-caspase-1, released by the formation of inflammasomes, is activated to cleave and mature pro-IL-1β. Additionally, IL-1β induces a substantial neuroinflammatory signal that alerts adjacent cells and eventually reaches the trigeminal nervous system around the meningeal vessels, activating this nervous system and resulting in a headache. In the CNS, CSD can induce the production of IL-1β by neurons, astrocytes, and microglia. IL-1β binds to the type I IL-1 receptor and activates the transcription factor NF-κB, which promotes the production of pro-inflammatory cytokines and generates an inflammatory cascade. Anakinra antagonizes the effects of IL-1β by inhibiting IL-1R1, and MCC950 stimulates the blockade of these neuroinflammatory cascades by inhibiting NLRP3, which might suppress the headache.

## 8. Conclusions

In this review, we present evidence for the substantive role of neuroinflammation in migraine. There is increasing evidence of abnormal cytokine production in both human and experimental models, and the inhibition of neuroinflammation may contribute to the improvement of migraine pathology [67,68]. Based on the described reviews, the Il-1β/IL-1R1 axis in particular plays a crucial role in both the peripheral TG [100,103,124] and trigeminovascular pathway in the CNS [69]. Thus, regulation of the IL-1β cascade by anakinra and MCC950 may lead to new therapeutic options for migraine.

Although these inflammatory mediators, including IL-1β, have been known to lead to BBB impairment, at present, BBB disturbances are indicated only in rodent models, while human studies have been negative. An elevation of inflammatory mediators, including MMP-9, and structural changes in pericytes, were detected in patients with migraine. Disruption of the BBB could not be completely ruled out, and stabilization of the BBB might be a potential therapeutic target for migraine in the future.

## Figures and Tables

**Figure 1 ijms-22-08929-f001:**
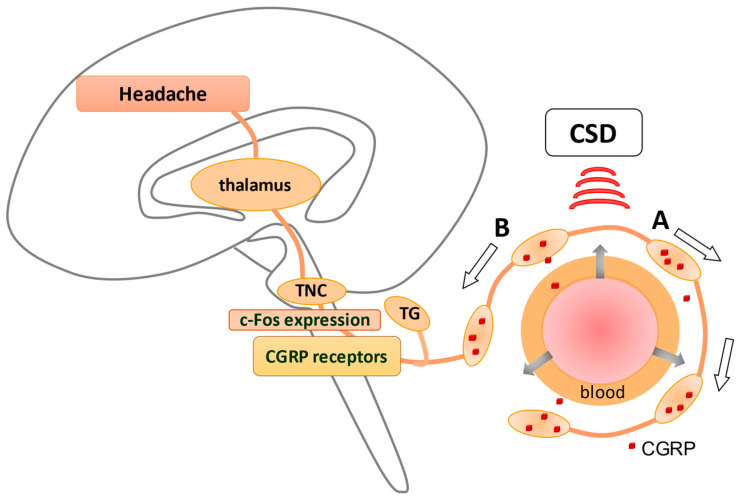
CGRP and the trigemino-vascular system in migraine.

**Figure 2 ijms-22-08929-f002:**
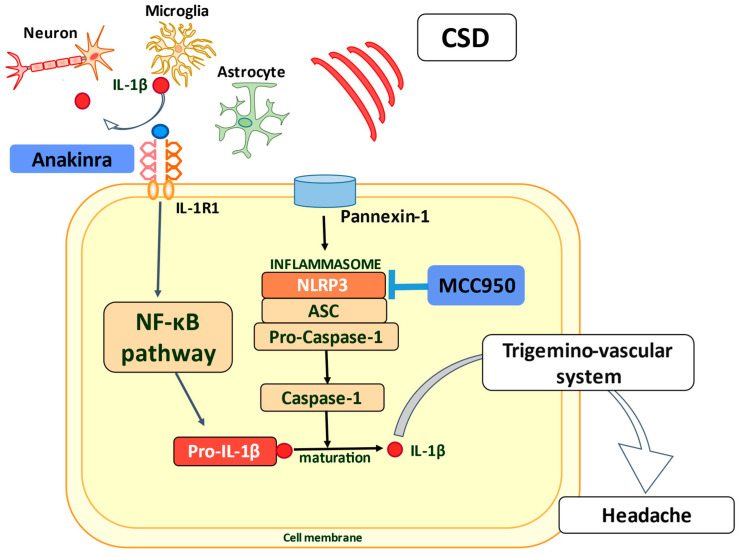
Potential for therapies using the IL-1β/IL-1R1 axis.

## Data Availability

The datasets generated and/or analyzed during the current study are available at the PubMed database repository (https://pubmed.ncbi.nlm.nih.gov/ (access date: 1 June 2021)).

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
