# Peer review of "Role of Neuroinflammation and Blood-Brain Barrier Permutability on Migraine"

_ijms, 2021, doi:10.3390/ijms22168929_

Round 1

Reviewer 1 Report

The current Review by Yamanaka and colleagues is of interest for the migraine field. Overall, the review is well-structured and well-written. However, there are some issues that need to be corrected in the revised version of the manuscript before it can be considered for publication in IJMS.

Line 197: Dietary polyphenol curcumin has been shown to significantly reduce the expression of IL-1beta, and TNF-alpha, oxidative stress, protein accumulation, and ultimately protect neuronal cells against neurodegeneration in several different models of disease (PMID: 17273796; PMID: 27197872; PMID: 30875761; PMID: 30666189). In addition, curcumin has been found to have preemptive analgesic and antioxidative effects on experimental migraine (PMID: 29204441). The authors should cite these studies, and briefly comment on curcumin’s potential beneficial effect on migraine-associated IL-1beta signalling pathway.

Line 280: The neuroprotective role of Anakinra on Interleukin-1 signaling pathway as also been shown in peripheral transthyretin amyloidosis (PMID: 25668561; PMID: 24918964). There studies are of relevance for current Review, and should be cited by the authors in the corrected version of the manuscript.

Author Response

Manuscript ID: ijms-1331043

Type of manuscript: Review

Title: Role of neuroinflammation and blood-brain-barrier permutability on

migraine

To the Reviewer 1

We appreciate the time and effort that the reviewer has dedicated to providing insightful feedback. We have revised our manuscript according to your comments as much as possible. All revisions and additions are indicated as underlined and yellow-highlighted text in the manuscript. We sincerely hope that with these revisions, our manuscript will be suitable for publication in the International Journal of Molecular Sciences.

Reviewer: 1
Comments

Referee1

The current Review by Yamanaka and colleagues is of interest for the migraine field. Overall, the review is well-structured and well-written. However, there are some issues that need to be corrected in the revised version of the manuscript before it can be considered for publication in IJMS.

Line 197: Dietary polyphenol curcumin has been shown to significantly reduce the expression of IL-1beta, and TNF-alpha, oxidative stress, protein accumulation, and ultimately protect neuronal cells against neurodegeneration in several different models of disease (PMID: 17273796; PMID: 27197872; PMID: 30875761; PMID: 30666189). In addition, curcumin has been found to have preemptive analgesic and antioxidative effects on experimental migraine (PMID: 29204441). The authors should cite these studies, and briefly comment on curcumin’s potential beneficial effect on migraine-associated IL-1beta signalling pathway.

Response: The following has been added to Lines 326-330

Recently, curcumin, a dietary polyphenol, has been demonstrated to significantly inhibit IL-1b and TNF-a expression, oxidative stress, and protein accumulation and to eventually protect neurons from neurodegeneration in several different disease models [119-122]. Additionally, curcumin has been shown to have a preemptive analgesic and antioxidant effect on experimental migraine [123].

Line 280: The neuroprotective role of Anakinra on Interleukin-1 signaling pathway as also been shown in peripheral transthyretin amyloidosis (PMID: 25668561; PMID: 24918964). There studies are of relevance for current Review, and should be cited by the authors in the corrected version of the manuscript.

Response: The following has been added to Lines 316-320

The inhibitory effect of anakinra on neuroinflammation has been reported not only in the CNS but also in the peripheral nervous system. In mouse models of familial amyloid polyneuropathy, anakinra inhibits transthyretin expression in Schwann cells [116] and extracellular deposition and protects against unmyelinated fibrillary degeneration [117].

Reviewer 2 Report

Yamanaka et al. review an involvement of pro-inflammatory cytokines such as IL-1β in migraine and suggest that the IL-1β/IL-1R1 axis may be a target for treating migraine, the idea of which is shown as a schematic diagram in Fig. 1. The authors also discuss whether migraine is due to blood-brain barrier impairment. This review article may be useful to migraine researchers who are interested in neuroinflammation. In order for non-experts to understand this review, it would be good to add explanations to some technical words used in the text. There are many points that should be addressed and may serve to amend this manuscript, as follows:

Major points:

  1. Abstract: it is not clear how “the change in the function of tight junction in animal models but not human patients” as mentioned in lines 22-24 on page 1 is related to the conclusion in Abstract. Please amend this point.
  2. Introduction: the authors should mention shortly an involvement of serotonin in the pathogenesis of migraine and a silver bullet for migraine, triptan, although these may be classical.
  3. Figure 1: the authors should use either IL-1R or IL-1 receptor throughout Fig. 1 and its legend. Does the precursor of IL-1β (line 301) mean Pro-IL-1β (Fig. 1)?; please make this point clear. In the cell expressing IL-1R1, is IL-1β produced, as shown in Fig. 1? How does IL-1β stimulate the trigemino-vascular system? What kind of cell is shown in Fig. 1? What neuronal region is the cell located in? Please make these points clear.
  4. The authors should give a diagram showing how the trigemino-vasucular system causes headache, including the expression of c-Fos and CGRP in the trigeminal nucleus caudalis (lines 290-292). Such a figure may serve to understand why migraine is currently treated by targeting CGRP (lines 17, 35-36 and 41-45).
  5. Many abbreviations are used in this manuscript. It would be nice to give a list of abbreviations to help the reader understand this manuscript.

Specific points:

  1. Line 36: is “since” OK?
  2. Line 71: not “IL1-β” but “IL-1β”.
  3. Line 74: an explanation of IL-10 should be given here but not in line 78.
  4. Line 75: please give a short explanation of sumatriptan.
  5. Lines 77 and 88: is MIP-1 either MIP-1α or MIP-1β?
  6. Lines 78 and 79: please correct English.
  7. Line 82: please explain shortly “sTNF-R1” and “s-TNF-R2”.
  8. Line 98: please explain shortly sICAM-1.
  9. Line 101: please explain shortly MCP-1.
  10. Line 103: please explain shortly selectin.
  11. Line 111: the authors should use IL-1ra or IL-1RA (see line 271) through the text; not “TGF-1b1” but “TGF-1β1”. Please explain shortly TGF-1β1.
  12. Line 129: please explain shortly “IL10RA” and “CSF1R”.
  13. Line 133: NTG should be defined in this line but not in line 174.
  14. Line 140: not “nociception” but “nociceptor”?
  15. Line 160: not “… molecules ICAM-1 …” but “… molecules (ICAM-1 …”?
  16. Line 161: not “IL-1 receptor-1” but “IL-1 receptor type 1” (line 266)?
  17. Line 164: not “agents” but “agent”?
  18. Line 184: “regulator” is not “injection” but “KYNA-A1”; please correct this line.
  19. Line 190: please explain shortly NLRP3.
  20. Line 192: “+” in “K+” should be superscript.
  21. Line 216: is “… the induced …” OK? Please correct English.
  22. Line 225: please explain shortly ESM-1; not “by” but “in”?
  23. Line 261: what kinds of antiepileptics? Please reply to this question.
  24. Line 268: please expand TG.
  25. Line 271: not “IL1β1” but “IL-1β1”?
  26. Line 276: not “exhibit” but “exhibits”.
  27. Line 300: please explain shortly ASC.
  28. There appear to be much more mistakes than pointed out above. The authors should check the manuscript very carefully.

Author Response

Authors' responses to reviewer's comments are attached, please check.

Reviewer 3 Report

Dear Authors, thanks for this interesting manuscript to review. 

I consider than introduction explain key concepts such as migraine and neurogenic inflammation. I consider that this information is useful for novel readers in migraine.

Authors develop a detailed narrative review of clinical and experimental evidence of neuroinflamation. Referes are updated and appropiated.

Related to presence of blood-brain barrier permeability the analysis is clear according previous evidence.

I thank the authors the Figure 1. 

The conclusion is according to the review design.

Discussion section is not necessary, because this is a report of previous evidence. 

Author Response

Manuscript ID: ijms-1331043

Type of manuscript: Review

Title: Role of neuroinflammation and blood-brain-barrier permutability on

migraine

To the Reviewer 1

We appreciate the time and effort that the reviewer has dedicated to providing insightful feedback. We sincerely hope that with these revisions, our manuscript will be suitable for publication in the International Journal of Molecular Sciences.

Comments

Referee3

Dear Authors, thanks for this interesting manuscript to review. 

I consider than introduction explain key concepts such as migraine and neurogenic inflammation. I consider that this information is useful for novel readers in migraine.

Authors develop a detailed narrative review of clinical and experimental evidence of neuroinflamation. Referes are updated and appropiated.

Related to presence of blood-brain barrier permeability the analysis is clear according previous evidence.

I thank the authors the Figure 1. 

The conclusion is according to the review design.

Discussion section is not necessary, because this is a report of previous evidence. 

Round 2

Reviewer 1 Report

The authors have successfully addressed my concerns, therefore I now fully recommend the manuscript for publication.

Author Response

Manuscript ID: ijms-1331043

Type of manuscript: Review

Title: Role of neuroinflammation and blood-brain-barrier permutability on

migraine

To the Reviewer 1

We appreciate the time and effort that the reviewer has dedicated to providing insightful feedback. We sincerely hope that with these revisions, our manuscript will be suitable for publication in the International Journal of Molecular Sciences.

Reviewer 2 Report

This revised version has been largely amended according to my comments, and there is no concern in the present manuscript except for the following minor comments:

  1. Line 47: not “Trigemino-vascular System in Migraine” but “trigemino-vascular system in migraine”.
  2. Line 96: not “sumatriptan administration, one of the triptans,” but “sumatriptan (one of the triptans) administration”.
  3. Line 105: what is [xx]?
  4. Line 153: not “interleukin 10 receptor” but “IL-10 receptor”?
  5. Line 297: not “IL1β1” but “IL-1β1”?
  6. Line 300: not “[103][11]” but “[11,103]”?
  7. Lines 325 and 335: not “Figure 1” but “Figure 2”.
  8. Line 338: not “neuron cells” but “neurons”?
  9. Line 376: not “metalloproteinases” but “metalloproteinase”.

Author Response

Manuscript ID: ijms-1331043

Type of manuscript: Review

Title: Role of neuroinflammation and blood-brain-barrier permutability on

migraine

To the Reviewer 2

We appreciate the time and effort that the reviewer has dedicated to providing insightful feedback. We have revised our manuscript according to your comments as much as possible. All revisions and additions are indicated as underlined and yellow-highlighted text in the manuscript. We sincerely hope that with these revisions, our manuscript will be suitable for publication in the International Journal of Molecular Sciences.

Response: The following items have been corrected as indicated. Finally, we would like to thank you for your valuable comments.

  1. Line 47: not “Trigemino-vascular System in Migraine” but “trigemino-vascular system in migraine”.
  2. Line 96: not “sumatriptan administration, one of the triptans,” but “sumatriptan (one of the triptans) administration”.
  3. Line 105: what is [xx]? delete
  4. Line 153: not “interleukin 10 receptor” but “IL-10 receptor”?
  5. Line 297: not “IL1β1” but “IL-1β1”?
  6. Line 300: not “[103][11]” but “[11,103]”?
  7. Lines 325 and 335: not “Figure 1” but “Figure 2”.
  8. Line 338: not “neuron cells” but “neurons”?
  9. Line 376: not “metalloproteinases” but “metalloproteinase”.